# Prevalence of SARS-CoV-2 Infection at the University of Barcelona during the Third COVID-19 Pandemic Wave in Spain

**DOI:** 10.3390/ijerph18126526

**Published:** 2021-06-17

**Authors:** Sebastián Videla, Aurema Otero, Sara Martí, M. Ángeles Domínguez, Nuria Fabrellas, M. Pilar Delgado-Hito, Imma Cruz, Cristian Tebé, Teresa Vinuesa, Fernando Ardila, Marta Sancho, Esteve Fernández, Montserrat Figuerola, Francisco Ciruela

**Affiliations:** 1Pharmacology Unit, Department of Pathology and Experimental Therapeutics, School of Medicine and Health Sciences, IDIBELL, University of Barcelona, L’Hospitalet de Llobregat, 08907 Barcelona, Spain; 2Clinical Research Support Unit (HUB-IDIBELL), Clinical Pharmacology Department, Bellvitge University Hospital, L’Hospitalet del Llobregat, 08907 Barcelona, Spain; auremaotero@gmail.com; 3Microbiology Unit, Department of Pathology and Experimental Therapeutics, School of Medicine and Health Sciences, IDIBELL, University of Barcelona, L’Hospitalet de Llobregat, 08907 Barcelona, Spain; smartinm@bellvitgehospital.cat (S.M.); adominguez@bellvitgehospital.cat (M.Á.D.); tvinuesa@ub.edu (T.V.); 4Microbiology Department, Bellvitge University Hospital, L’Hospitalet del Llobregat, 08907 Barcelona, Spain; 5Consortium for Biomedical Research in Respiratory Diseases (CIBERES), 28029 Madrid, Spain; 6Department of Public Health Nursing, Mental Health and Maternal and Childhood, School of Medicine and Health Sciences, University of Barcelona, 08036 Barcelona, Spain; nfabrellas@ub.edu; 7Department of Fundamental and Medical-Surgical Nursing, School of Medicine and Health Sciences, GRIN-IDIBELL, University of Barcelona, L’Hospitalet de Llobregat, 08907 Barcelona, Spain; pdelgado@ub.edu; 8OSSMA (Oficina de Seguretat, Salut i Medi Ambient), University of Barcelona, 08028 Barcelona, Spain; imma.cruz@ub.edu; 9Biostatistical Unit, Department of Clinical Sciences, School of Medicine and Health Sciences, IDIBELL, University of Barcelona, L’Hospitalet de Llobregat, 08907 Barcelona, Spain; ctebe@idibell.cat; 10UICEC-IDIBELL (Clinical Research Organization), L’Hospitalet de Llobregat, 08907 Barcelona, Spain; fardila@bellvitgehospital.cat; 11Gerència Territorial Metropolitana Sud, Catalan Institute of Health, L’Hospitalet de Llobregat, 08907 Barcelona, Spain; msancho@ambitcp.catsalut.net (M.S.); m.figuerola@metrosud.cat (M.F.); 12Department of Cancer Epidemiology and Prevention, Institut Català d’Oncologia—ICO, L’Hospitalet de Llobregat, 08908 Barcelona, Spain; 13Program of Epidemiology and Public Health, Institut d’Investigació Biomèdica de Bellvitge, L’Hospitalet de Llobregat, 08908 Barcelona, Spain; 14Department of Clinical Sciences, School of Medicine and Clinical Sciences, University of Barcelona, L’Hospitalet de Llobregat, 08907 Barcelona, Spain

**Keywords:** coronavirus, seroprevalence, SARS-CoV-2, infection status, university community, COVID-19 prevalence, faculty members, Spain, students, administrative and service staff

## Abstract

The severe acute respiratory syndrome coronavirus 2 (SARS-CoV-2) pandemic started in December 2019 and still is a major global health challenge. Lockdown measures and social distancing sparked a global shift towards online learning, which deeply impacted universities’ daily life, and the University of Barcelona (UB) was not an exception. Accordingly, we aimed to determine the impact of the SARS-CoV-2 pandemic at the UB. To that end, we performed a cross-sectional study on a sample of 2784 UB members (*n* = 52,529). Participants answered a brief, ad hoc, online epidemiological questionnaire and provided a nasal swab for reverse transcription polymerase chain reaction (RT-PCR) SARS-CoV-2 analysis and a venous blood sample for SARS-CoV-2 IgG antibody assay. Total prevalence of SARS-CoV-2 infection (positive RT-PCR or positive IgG) was 14.9% (95%CI 13.3 to 17.0%). Forty-four participants (1.6%, 95%CI: 1.2–2.1%) were positive for SARS-CoV-2 RT-PCR. IgG against SARS-CoV-2 was observed in 12.8% (95%CI: 11.6–14.1%) of participants. Overall, while waiting for population vaccination and/or increased herd immunity, we should concentrate on identifying and isolating new cases and their contacts.

## 1. Introduction

The severe acute respiratory syndrome coronavirus 2 (SARS-CoV-2) pandemic remains a major global health challenge which limits universities’ academic and research activities, thereby forcing the implementation of new teaching and working paradigms (online distance learning, etc.). Interestingly, SARS-CoV-2 transmission at the university has been associated with the well documented infectivity of asymptomatic individuals, many of them being pre-symptomatic with high viral loads [1,2,3,4,5,6]. Those asymptomatic carriers are likely to be responsible for as many as 44% of new infections [7]. Although closing campuses and switching to fully remote education will reduce SARS-CoV-2 transmission, this measure of force might have impacts on education quality, graduation rates and revenue [8].

Indeed, the COVID-19 pandemic has changed university life. On-site activities have been reduced only to those considered essential and requiring face-to-face attendance (i.e., demonstrations). While these challenges posed by the pandemic have created unique communication opportunities for university communities, it must be recognized that, for many courses (e.g., organic chemistry), it is challenging to complete an academic year taught exclusively online, and that professional and academic skills training have been jeopardized by the pandemic. In addition, online administrative procedures and telecommuting means had to be provided to university employees, which meant a considerable investment in a time of economic turmoil. Within this scenario, the main concerns are the impacts that this temporary closure may have on the quality of education and the academic performance of students. Overall, while it is important to maintain students’ academic performance during this crisis period [9], public health cannot be compromised by an in loco education system favoring SARS-CoV-2 transmission; thus, many considerations should be balanced before deciding to reopen the university.

The University of Barcelona (UB) is a public Catalan university founded in 1450 and is the largest university in Catalonia. It is frequently ranked as the first-rated university in the country [10]. During the last academic year (2019/20) the UB community was comprised of 69,353 members, including 61,119 students (~17% foreign) distributed as 41,750 bachelor’s degree students, 5337 graduate students, 4582 pre-doctoral researchers and 8941 postgraduate students [11]. In addition, the UB employs 2409 administrative and service staff (ASS), and 5825 faculty members (FM), of which 930 are clinical faculty members (CFM) associated with the three main university hospitals (Clínic, Bellvitge and Sant Joan de Déu) and 11 corporate health centers [12] of the UB. Altogether, UB members are distributed all through an institutional structure consisting of 16 schools organized into six university Campuses [13], the Barcelona Science Park [14], 17 UB research institutes and other 12 research institutes [15].

Antibodies are a biomarker for total or partial immunity, so their prevalence can reveal the proportion of the population that remains susceptible to the virus. Knowing the prevalence of infection (active or passed) constitutes a powerful tool for overviewing the pandemic’s impact at the UB to guide academic authorities toward the re-opening of on-site activities progressively.

## 2. Materials and Methods

### 2.1. Study Design and Participants

This was a cross-sectional study conducted among UB members between 14 December 2020 and 25 February 2021. According to a nationwide, population-based seroepidemiological study (ENE-COVID Study) [16], we expected a SARS-CoV-2 seroprevalence of 7.5% for students and 12% or higher for clinical faculty members. In addition, we assumed a 30% non-response rate. Thus, to reach a final overall sample size of 3450 individuals, we estimated that an initial sample size of 4944 participants was required. On 9 December 2020, 4944 UB members were randomly selected and invited to participate by email. After three reminders, only 370 individuals participated in the study, thereby providing rate of participation of 7.5% (almost ten-times less to the expected one). Consequently, on 1 February 2021, after this initial attempt, a new sample of 4944 UB members was randomly selected and invited to participate by email with similar results. Finally, in view of this low participation rate, we decided to contact and invite the remaining UB population to participate in two other successive waves, on 8 February (*n* = 26,671) and 15 February 2021 (*n* = 15,961). A final sample of 2784 participants participated (Figure 1). UB members were contacted by the information in the most recent census (updated for the UB President’s election in December 2020).

### 2.2. Logistics Procedure

The email briefly introduced the study and requested participation, which entailed free PCR and IgG testing. Once a participant accepted online, he/she was required to answer a short online epidemiological questionnaire. This questionnaire (see Table A1 in Appendix A) gathered information about sociodemographic variables, self-reported clinical background (including estimated body masa index and COVID-19-related symptoms), lifestyle habits (i.e., tobacco and alcohol use), previous screening for SARS-CoV-2 (i.e., RT-PCR and/or serology) and risk of SARS-CoV-2 infection (i.e., contact with infected people). Thereafter, the participant was able to choose the day and the hour for sample collection in one of the three UB points of care in the city—two at the UB Medical School campuses (Clínic and Bellvitge) and one at UB Health Services (Pedralbes Campus). Next, the participant received an email with the appointment. If needed, the participant was able to amend the appointment with the support of the study personnel.

### 2.3. Sample Collection

Participants present at the UB points of care were first asked to sign the written informed consent to participate in the study and review the online epidemiological questionnaire with an interviewer of the study team. Thereafter, trained nurses obtained a nasal sample with a mid-turbinate swab for RT-PCR testing [17] and a venous blood sample (3 mL) for detection of SARS-CoV-2 antibodies. Samples were assigned numeric codes for de-identification purposes and were processed by the Microbiology Service of the Bellvitge University Hospital. When a positive RT-PCR result was found, the participant was immediately contacted and referred to the COVID-19 agent from the Catalan Health Service, thereby following the established COVID-19 protocol.

### 2.4. SARS-CoV-2 Detection by RT-PCR

SARS-CoV-2 active infection was studied on mid-turbinate nasal swabs by RT-PCR using the TaqPathTM^®^ COVID-19 assay (Thermo Fisher Scientific, Madrid, Spain). Values below 40 cycles were taken as positive results for SARS-CoV-2. Presumptive identification of cases belonging to the variant of concern (VOC) 202012/01 (B.1.177 lineage) [18] was assessed by TaqPathTM^®^ when both viral targets ORF1ab and N yielded positive amplifications while the S target provided a negative result [19].

### 2.5. Detection of SARS-CoV-2 Antibodies

Detection of SARS-CoV-2 antibodies in serum samples was carried out by the Elecsys^®^ Anti-SARS-CoV-2 electrochemiluminescence immunoassay (Roche Diagnostics GmbH, Mannheim, Germany), used for the in vitro qualitative detection of antibodies (including IgG) against SARS-CoV-2 in human serum and plasma. The assay uses a recombinant protein representing the nucleocapsid (N) antigen in a double-antigen sandwich assay format, which favors detection of high affinity antibodies against SARS-CoV-2. Elecsys^®^ Anti-SARS-CoV-2 detects antibody titers, which have been shown to positively correlate with neutralizing antibodies in neutralization assays [20,21].

### 2.6. Statistical Analysis

Participants in our study were randomly selected through stratified one-stage sampling from the entire UB population. Due to the heterogeneity of sociodemographic characteristics across the UB population, stratification was based on students, ASS and faculty members. This last group was also divided into clinical faculty and non-clinical faculty members (i.e., CFM and FM) due to an expected higher exposition to SARS-CoV-2 among the first. By using this four-group stratification, no UB member was left out of the study. The sample size by the group was determined for an underlying SARS-CoV-2 seroprevalence of 7.5% or higher for students, ASS, FM and CFM, according to a nationwide, population-based seroepidemiological study (ENE-COVID Study) [16], and 12% or higher for clinical faculty.

Baseline characteristics of participants by group (i.e., students, ASS, FM and CFM) are described using mean and standard deviation for continuous variables and frequencies for categorical variables. Prevalence of asymptomatic SARS-CoV-2 infection is reported as a percentage of subjects with a positive RT-PCR. Seroprevalence was estimated as the percentage of subjects with a positive serology test. Global RT-PCR-positive prevalence and global gseroprevalence were estimated using sampling weights. Exact 95% binomial confidence intervals were calculated for each prevalence. For sensitivity, prevalence of asymptomatic SARS-CoV-2 infection and seroprevalence were estimated by recruitment period, and by health-related faculty (i.e., medicine, biology, psychology and pharmacy). Data analysis was carried out using R statistical software [22].

### 2.7. Role of the Funding Source

The funders had no role in the design, analysis, interpretation or writing. The first three authors (S.V., A.O. and E.F.) and the senior author (F.C.) had full access to all the data, and they had final control over the decision to submit this paper for publication.

## 3. Results

### 3.1. Baseline Characteristics

From the UB community members invited to participate via email (52,529 members), only 3243 enrolled into the study, which represents 6.2% of the UB population invited (Figure 1). The list of enrolled UB participants was then refined by searching redundancy (duplicities, etc.) and/or information mismatches; thus, we ended with a study population of 3123 UB members with a complete epidemiological questionnaire. From these, we were unable to obtain biological samples (i.e., nasal swap and blood sample) from 339 individuals, thereby ending in a final study population of 2784 participants providing both a validated epidemiological questionnaire and SARS-CoV-2 analytical result (Figure 1).

The sample (*n* = 2784) was constituted by 1206 graduate and undergraduate students, 699 ASSs, 793 FMs and 86 CFMs (Table 1), with a high proportion of women (65.3%). The mean age of the participating students was 23.1 (SD, 6.3) years old; the mean age of ASS, FM and CFM was 49.1 (SD, 0.4) years old (Table 1), which is near that reported in the UB records (i.e., 48.5 ± 1.2 years old). Baseline characteristics of the participants by university group are shown in Table 1. Interestingly, 54% of participants declared daily alcohol consumption and 13% were current smokers, and less than 11% and 7% declared having increased use during the pandemic, respectively (see Table 1).

From participants that declared to have been previously tested for SARS-CoV-2 by RT-PCR (44%) or serology analysis (16%), 10% of them reported a positive RT-PCR and 13% informed us about positive SARS-CoV-2 serology (see Table 1 for detailed stratification). Interestingly, 16% of participants declared being in direct contact with infected people (Table 1). Among the 2784 participants previously tested for SARS-CoV-2 with at least one RT-PCR (Table 1), 1225 (44.0%, 95%CI: 42.2–45.9%) provided information about COVID-19-related symptoms and signs (Table 2); thus, 360 (29.4%) reported at least one COVID-19-related symptom. The more frequent symptoms were fever (14.4%) and cough (11.6%); the less frequent were ageusia (2.78%) and nausea (3.26%) (Table 2).

### 3.2. SARS-CoV-2 Infection Prevalence

Total prevalence of SARS-CoV-2 infection (positive RT-PCR or positive IgG) was 14.9% (95%CI 13.3 to 17.0%). The prevalence of SARS-CoV-2 infection determined by RT-PCR (active infection) was 1.59% (44 out of 2775, 95%CI: 1.18–2.12%). No differences arose by groups within the UB community. Students had the highest prevalence (2.08%, 95%CI: 1.35–3.06%). ASS and FM showed prevalences of 1.0% (95%CI: 0.40–2.06%) and 1.52% (95%CI: 0.79–2.64%), respectively. There were no SARS-CoV-2 RT-PCR-positive participants among the CFM. Finally, no differences in active infection were found by sex (1.03%, 95%CI: 1.54–2.22% female; and 0.95%, 95%CI: 1.66–2.69% male). Global SARS-CoV-2 infection prevalence estimated using sampling weights and SARS-CoV-2 infection prevalence by UB community groups are shown in Figure 2.

At the time of the study, none of the participants were yet vaccinated; thus, the N antigen was enough to detect immunity after SARS-CoV-2 infection. The overall prevalence of past SARS-CoV-2 infection was 12.8% (356 out of 2775, 95%CI: 11.6–14.1%). The seroprevalence among students was higher (15.4%, 95%CI: 13.4–17.6%) and statistically significant compared to FM (9.0%, 95%CI: 7.1–11.2%; *p* < 0.05). The seroprevalence among ASS was 12.9% (95%CI: 10.5–15.6%), and it was 11.8% (95%CI: 5.8–20.6%) among CFM. By sex, the prevalence of past SARS-CoV-2 infection was 11.66%, 95%CI: 13.19–14.84%, in females; and 10.15%, 95%CI: 12.15–14.38%, in males. Global seroprevalence estimated using sampling weights and seroprevalence by UB community groups are shown in Figure 2.

Importantly, while the 44 participants with positive SARS-CoV-2 RT-PCR assays were clinically asymptomatic, six out of 44 showed the presumptive VOC 202012/01 genotype, first described in UK in early December 2020 [19]. In two out of these six cases, antibody detection was positive, and it was negative in the other four cases. All six participants were detected in February 2021, a period of greater detection of this viral variant in our area.

### 3.3. Asymptomatic COVID Infections

From the 44 asymptomatic participants who were RT-PCR-positive for SARS-CoV-2, 38 (86%) also carried IgG antibodies. Therefore, only six participants (four students and two FMs) were considered to have early infections (RT-PCR-positive but negative serological assays). These individuals constituted the 0.22% (6 out of 2775, 95%CI: 0.10–0.47%), likely considered as asymptomatic cases with high viral loads, and thus potential transmitters, as discussed earlier [1,2,3,4,5,6].

Interestingly, among participants who declared at least one COVID-19-related symptom (*n* = 360), the proportions of participants reporting a seropositive study were 29.2% (95%CI: 22.9–36.0%) among students; 30.1% (95%CI: 20.5–41.2%) among ASS, 25.7% (95%CI: 16.0–37.6%) among FC and 37.5% (just 3 out of 8, 95%CI: 8.5–75.51%) among CFM. By sex, 27.4%, 95%CI: 22.1–33.3% (71 out of 259) were female; 32.7%, 95%CI: 23.7–42.7% (33 out of 101) were male. From those, positive RT-PCR tests were reported by only 10 participants (five males and five females).

## 4. Discussion

The findings from this study (carried out mainly in February 2021, during the pandemic’s third wave in Spain) indicate a relatively low prevalence of SARS-CoV-2 infection within the UB community. We did not observe substantial differences in SARS-CoV-2 infection by community group or sex, other than between students and FMs. All participants who had positive RT-PCR tests were asymptomatic. However, as time goes by and SARS-CoV-2 infections spread, this low prevalence will grow, and together with community vaccination will contribute to herd immunity. Accordingly, a close follow up of the pandemic’s progression within the next months will allow us to progressively adapt the teaching and research activities towards normality.

Asymptomatic and paucymptomatic individuals can unknowingly transmit the virus and fuel covert outbreaks [5,23,24]. Our results show that the prevalence of asymptomatic SARS-CoV-2 infected individuals (1.94%) was low, as expected, but this finding does not guarantee safety. Our results cannot rule out that close contact with people with COVID-19, and particularly those in the same household, increases viral transmission. The early detection of asymptomatic infections is vital for mitigating viral transmission and containing outbreaks. Therefore, the information about asymptomatic SARS-CoV-2 infected individuals is essential for guiding university directives regarding re-opening on-site activities.

Serological screening is the best tool to determine the spread of an infectious disease, particularly in the presence of asymptomatic cases or incomplete ascertainment of those with symptoms [16,25]. The seroprevalence found (12.8%) was greater than that estimated in Catalonia in December 2020, 9.2% (95%CI: 7.7–11.0)—by gender, 8.6% (95%CI: 6.9–10.5) in males and 9.8% (95%CI: 8.1–11.8) in females [16]; and greater than the seroprevalence estimated in Spain in the same study (overall population, 7.1% (95%CI: 6.7–7.6), and by gender, 6.7% (95%CI: 6.2–7.2) in males and 7.5% (95%CI: 6.9–8.0) in females). Interestingly, the results reported here are in line with these recently reported for the SARS-CoV-2 seroprevalence in 2905 university students from five different universities within England (17.8%, 95%CI, 16.5–19.3), ranging between 7.6 and 29.7% [26].

Detection of SARS-CoV-2 infection performed simultaneously by RT-PCR and IgG serology provides a global view of the pandemic’s impact within the university community. Asymptomatic SARS-CoV-2 infected people cannot be ignored, and the low seroprevalence found is plainly insufficient to be considered as satisfactory herd immunity. Considering the global situation (including economic recession) and while we wait for most of the population to be vaccinated and/or for herd immunity to be achieved, our results should be interpreted as sufficient to support the current nonpharmacologic preventive measures taken, such as mask wearing, social distancing, hand hygiene (with hydroalcoholic gels), ventilated spaces and “sheltering in place” for a minimum of 10 days if in contact with an infected person. Extensive social distancing with a mandatory mask-wearing policy can prevent most COVID-19 cases on college campuses and is very cost-effective; and routine laboratory testing would prevent disease spread, but would require lower-cost tests combined with markedly increased capacity to be feasible [27]. However, given the lack of logistical and economic competences for implementing screening tests of SARS-CoV-2 infection, all this means that the university officers will face the challenge soon of slowly returning to pre-pandemic university activity.

Online distance learning should be carried on, and on-site activities in UB should be only considered for those deemed essential. Although the university has attempted to maintain normality for their students through online lectures and video conferences, and by getting involved in the delivery of telehealth services, FMs must try to adapt to the new reality of the online classes with the aim of maintaining the level of educational excellence. Furthermore, the student–teacher interaction online is the new reality and must be accepted. This pandemic situation should be understood as an opportunity to develop new learning methods. In the future, online classes will be integrated as another pedagogical tool, and future lectures will be implemented for students worldwide. Hence, online classes’ innovation requires implementing procedures that potentiate real-time interaction with students. This should be one of our goals. The university should facilitate the training of professional skills in this reality and provide the means to carry it out [28]. Telecommuting is a measure that should be kept for a percentage of staff. Therefore, the procedures for user interaction, that of both FMs and students, should be improved or implemented. This “new reality” of the rapid university education transformation due to the COVID-19 crisis should be understood as an opportunity for positive and sustained change. The university must integrate the digital transformation gained during the COVID-19 pandemic in future academic courses, and must ensure that it continues, instead of going back to the pre-COVID status quo [28]. Importantly, the UB is a research-intensive institution operating in fields, including experimental and health sciences. Thus, the COVID-19 pandemic also represented a challenge for these UB research activities requiring in-person attendance. Indeed, research activities that are deemed critical (for example, animal research) have been permitted for designated personnel at designated times, and always with strict adherence to social distancing and preventive measures. Subsequently, several measures were undertaken to maintain and progressively increase baseline research. Overall, the COVID-19 pandemic represents a great challenge for the development of both teaching and research activities at our university.

Some limitations need to be considered for the interpretation of the results of this cross-sectional study. Whilst we intended to obtain a representative sample of the university community, the low participation rate prompted us to finally invite all members of the target community to participate, thereby allowing us to obtain a sufficient sample size. The final sample obtained was representative across several baseline characteristics, such as sex and age, of the overall university population, but not of community groups and schools. This might imply that generalization to the overall UB population or more general university populations must be done with caution. Finally, other potential limitations of the study should be considered—for instance, the fact that the clinical background data shown in Table 1 and Table 2 are based on self-reporting by participants (i.e., not formally recorded by a clinician); thus, this information should be considered accordingly. In addition, our questionnaire did not record information regarding the preventive measures taken by participants during their daily tasks. Overall, the focus of the study was to provide a global perspective of the pandemic’s impact on the UB community. Thus, although we studied heterogeneity across the UB community groups (i.e., students, ASS, FM and CFM), we did not account for the heterogeneity of transmission within each group (for example, students living outside of Barcelona vs. students living in Barcelona), which would have required additional and unavailable data. This could be of importance, since the UB welcomes a high percentage of students from other Spanish cities and from abroad, who during the lock-down and permanent closing of the university mostly stayed home. However, this study is, to the best of our knowledge, the first to assess prevalence of SARS-CoV-2 infection at a public university in Spain, and one of the few in the international literature [26].

## 5. Conclusions

The estimated prevalence of SARS-CoV-2 infection was low within UB’s community at the time we performed the study (i.e., February 2021). While these results do not suffice to deescalate COVID-19 preventive measures (i.e., mask wearing, social distancing and ventilated spaces), right now, the continuous increase of SARS-CoV-2 infection prevalence and community vaccination will contribute to achieving the minimal herd immunity needed to start taking progressive measures to adapt the teaching and research activities towards normality along the next academic year.

## Figures and Tables

**Figure 1 ijerph-18-06526-f001:**
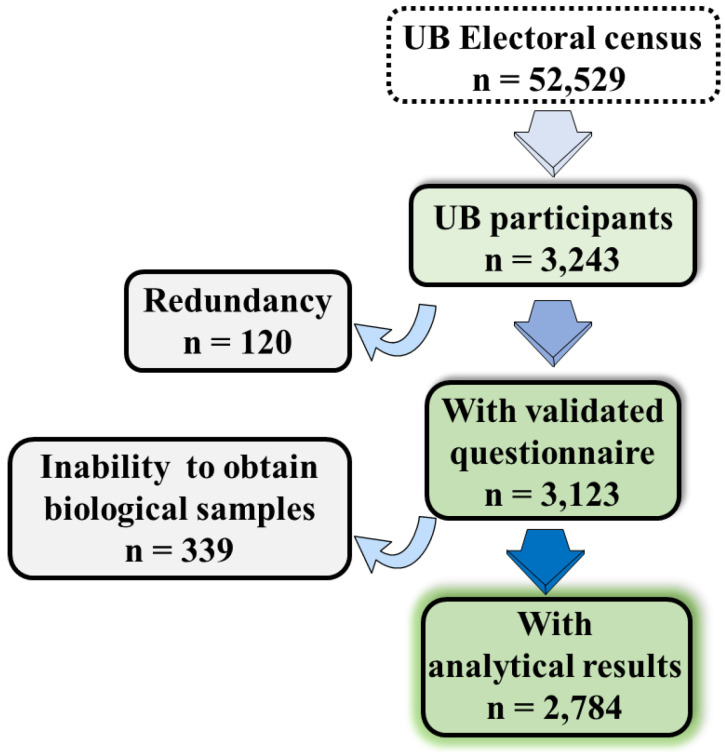
Overview of the flow chart of UB members involved in the study.

**Figure 2 ijerph-18-06526-f002:**
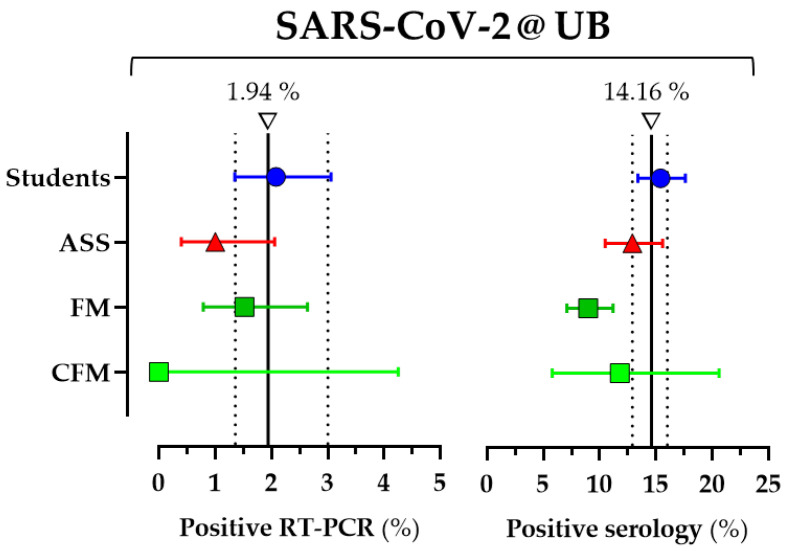
SARS-CoV-2 infection prevalence (assayed by RT-PCR and IgG serology) estimated using sampling weights by UB community groups. Results are expressed as percentages of subjects showing positive RT-PCR for SARS-CoV-2 (prevalence of current infection) or antibodies against SARS-CoV-2 (seroprevalence or prevalence of past infection) with 95% Cis, according to UB community groups: students, administrative and service staff (ASS); faculty members (FM) and clinical faculty members (CFM). A vertical line represents the overall prevalence (RT-PCR or serology) and the dotted lines the 95%CI.

**Table 1 ijerph-18-06526-t001:** Baseline characteristics of participants based on the epidemiological questionnaire.

		Students *n* = 1206	ASS *n* = 699	FM *n* = 793	CFM *n* = 86
**Age**, years	mean (sd)	23.1 (6.3)	49.3 (9.4)	48.8 (11.3)	50.4 (10.8)
**Gender**, Male/Female	n (%)/n (%)	334 (28)/872 (72)	222 (32)/477 (68)	369 (46.5)/424 (53.5)	40 (46.5)/46 (53.5)
**BMI**, kg/m²					
Underweight, < 18	n (%)	62 (5)	5 (1)	14 (2)	1 (1)
Normal weight, 18 < 25	n (%)	937 (78)	346 (50)	452 (57)	52 (61)
Overweight, 25 < 30	n (%)	172 (14)	238 (34)	266 (34)	27 (31)
Obesity, ≥ 30	n (%)	34 (3)	108 (16)	59 (7)	6 (7)
**Clinical background** (yes):	n (%)				
Cancer	n (%)	2 (0)	3 (0)	1 (0)	0 (0)
Cardiovascular disease (includes hypertension)	n (%)	9 (1)	73 (10)	89 (11)	13 (15)
Endocrine disease (diabetes)	n (%)	2 (0)	15 (2)	8 (1)	0 (0)
Immunocompromised	n (%)	3 (0)	12 (2)	9 (1)	1 (1)
Liver diseases	n (%)	2 (0)	6 (1)	2 (0)	0 (0)
Pulmonary disease	n (%)	38 (3)	44 (6)	26 (3)	2 (2)
Renal disease	n (%)	0 (0)	2 (0)	6 (1)	1 (1)
**Lifestyle habits** **:**					
Alcohol consumption, yes	n (%)	502 (42)	397 (57)	560 (71)	52 (61)
Increased during pandemic, yes	n(%)	57 (11)	47 (12)	59 (10)	8 (15)
Smoke tobacco, yes	n (%)	147 (12)	123 (18)	82 (10)	9 (11)
Increased during pandemic, yes	n (%)	71 (6)	47 (7)	35 (4)	1 (1)
**Previous screening for SARS-CoV-2** **:**					
At least one RT-PCR, yes	n (%)	626 (52)	263 (38)	292 (37)	44 (51)
At least one RT-PCR-positive, yes	n (%)	68 (11)	29 (11)	19 (7)	4 (9)
At least one serology study, yes	n (%)	232 (19)	78 (11)	126 (16)	16 (19)
At least one serology-positive study, yes	n (%)	35 (13)	15 (16)	12 (9)	4 (9)
**Risk of SARS-CoV-2 infection** **:**					
Direct contact with infected people, yes	n (%)	154 (23)	84 (13)	79 (12)	10 (13)

Abbreviations: ASS, administrative and service staff; FM, faculty members; CFM, clinical faculty members; BMI, body mass index.

**Table 2 ijerph-18-06526-t002:** COVID-19-related symptoms during 2020 in participants with at least one previous RT-PCR.

		Students *n* = 626	ASS *n* = 263	FM *n* = 292	CFM *n* = 44
Fever (yes)	n (%)	102 (16.3%)	38 (14.4%)	33 (11.3%)	4 (9.1%)
Cough (yes)	n (%)	77 (12.3%)	35 (13.3%)	26 (8.9%)	4 (9.1%)
Anosmia (yes)	n (%)	35 (5.6%)	10 (3.8%)	5 (1.7%)	1 (2.3%)
Ageusia (yes)	n (%)	26 (4.2%)	5 (1.9%)	2 (0.7%)	1 (2.3%)
Shortness of breath (yes)	n (%)	29 (4.6%)	8 (3.0%)	8 (2.7%)	3 (6.8%)
Sore throat (yes)	n (%)	54 (8.6%)	21 (8.0%)	12 (4.1%)	0 (0.0%)
Fatigue (yes)	n (%)	55 (8.8%)	20 (7.6%)	20 (6.8%)	2 (4.6%)
Nausea (yes)	n (%)	18 (2.9%)	11 (4.2%)	11 (3.8%)	0 (0.0%)
Diarrhoea (yes)	n (%)	50 (8.0%)	29 (11.0%)	15 (5.1%)	1 (2.3%)
Arthralgia (yes)	n (%)	56 (8.9%)	24 (9.1%)	12 (4.1%)	1 (2.3%)

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
