# Peer review of "Prevalence of SARS-CoV-2 Infection at the University of Barcelona during the Third COVID-19 Pandemic Wave in Spain"

_ijerph, 2021, doi:10.3390/ijerph18126526_

Round 1

Reviewer 1 Report

Thank you for the opportunity to review the manuscript.

The study by Videla et al describes a cross-sectional study that aimed to examine the prevalence of SARS-CoV-2 infection among students and staff at the University of Barcelona, Spain. To this end, the authors carried out an online epidemiological questionnaire, nasal swab for RT-PCR SARS-CoV-2 analysis, and blood sampling for SARS-CoV-2 IgG antibody. The total prevalence of SARS-CoV-2 infection (positive RT-PCR or positive IgG) was 14.9% and 1.6% were PCR positive.

The study reads well but lacks some issues that need to be addressed.

  1. The statistical analysis provided needs some revision, specifically analysis of variance (ANOVA) for comparing multiple means and comparing multiple proportions (chi-square) for Tables 1 and 2
  2. The clinical background data is based on self-reporting by participants and not medical data that was recorded or obtained by another party. This should be mentioned at least as a limitation of the study.
  3. The questionnaire itself focused on few items but did not address other important features such as adherence to mask-wearing and social distancing, average weekly hours that an individual spends indoors out of his home, etc. this should at least be addressed as a limitation.

Author Response

Thank you for the opportunity to review the manuscript.

The study by Videla et al describes a cross-sectional study that aimed to examine the prevalence of SARS-CoV-2 infection among students and staff at the University of Barcelona, Spain. To this end, the authors carried out an online epidemiological questionnaire, nasal swab for RT-PCR SARS-CoV-2 analysis, and blood sampling for SARS-CoV-2 IgG antibody. The total prevalence of SARS-CoV-2 infection (positive RT-PCR or positive IgG) was 14.9% and 1.6% were PCR positive.

The study reads well but lacks some issues that need to be addressed.

  1. The statistical analysis provided needs some revision, specifically analysis of variance (ANOVA) for comparing multiple means and comparing multiple proportions (chi-square) for Tables 1 and 2

Response:

The reviewer is right, a statistical analysis of the sociodemographic and clinical information provided in Table 1 and 2 might be interesting. However, the aim of the study presented here was to report the prevalence of SARS-CoV-2 infection at the University of Barcelona (UB) during the third COVID-19 pandemic wave. To this end, we performed a cross-sectional study among three different UB community groups. Thus, the only aim of these two tables is to describe the inherent sociodemographic and clinical characteristics of the subjects included in the study. Accordingly, assessing potential sociodemographic and clinical differences among groups is out of the scope of this study. Nevertheless, the STROBE statement discourages the use of p-values in descriptive tables in observational and cohort studies (Vandenbroucke et al. 2007 Ann Intern Med. 147: W-163–W-194).

  1. The clinical background data is based on self-reporting by participants and not medical data that was recorded or obtained by another party. This should be mentioned at least as a limitation of the study.

Response:

We agree with the referee that the clinical background data reported in Table 1 and 2 should be taken with caution as was self-reported by the participants. Thus, we highlight this as a potential limitation in the discussion section of the new version of the manuscript (“Finally, other potential limitations of the study should be considered, for instance the fact that the clinical background data shown in Table 1 and 2 is based on self-reporting by participants (i.e., not formally recorded by a clinician), thus this information should be considered accordingly”).

  1. The questionnaire itself focused on few items but did not address other important features such as adherence to mask-wearing and social distancing, average weekly hours that an individual spends indoors out of his home, etc. this should at least be addressed as a limitation.

Response:

The reviewer highlighted and important issue here. Our questionnaire did not recorded information regarding the preventive measures taken by participants during their daily tasks. Again, this might be considered a limitation of the study, thus we mentioned accordingly in the discussion section ("Finally, other potential limitations of the study should be considered, for instance ……. that our questionnaire did not recorded information regarding the preventive measures taken by participants during their daily tasks”).

Reviewer 2 Report

Dear Authors,

The manuscript is good, and well-written in most parts. The manuscript has sufficient originality, and undertaken problem is of practical nature. Although the results presented in the manuscript seem promising and overall approach is contributing to the body of the literature, I encourage the authors to please consider the attached file comments and suggestions to improvise the presented work more prior to its publication. Thanks

Author Response

Comments and Suggestions

1-          In abstract, the research question or motivation must be highlighted to emphasize the use of the proposed work. I would suggest the author to re-write the abstract in following way concisely.

[1]         What methods to use

[2]         What problems to solve

[3]         Experimental results analysis and discussion

[4]         Evaluation of the proposed method

Response: We thank the reviewer for his/her constructive comments. We followed the journal’s instructions for abstract completion (i.e., Single paragraph of 200 words max, without headings and organized as 1) Background: Place the question addressed; 2) Methods: Describe briefly the main methods. 3) Results: Summarize the article's main findings; and 4) Conclusion: Indicate the main conclusions or interpretations). Accordingly, we rewrite the abstract following these recommendations:

(1) Severe acute respiratory syndrome coronavirus 2 (SARS-CoV-2) pandemic started in December 2019 and still is a major global health challenge. Lockdown measures and social distancing sparked a global shift towards online learning, which deeply impacted universities’ daily life, and the University of Barcelona (UB) was not an exception. Accordingly, we aimed to determine the impact of SARS-CoV-2 pandemic at UB.

(2) To this end, we performed a cross-sectional study among a sample of 2,790 UB members (n = 52,529). Participants answered a brief ad hoc online epidemiological questionnaire and provided a nasal swab for reverse transcription polymerase chain reaction (RT-PCR) SARS-CoV-2 analysis and a venous blood sample for SARS-CoV-2 IgG antibody study.

(3) Total prevalence of SARS-CoV-2 infection (positive RT-PCR or positive IgG) was 14.9% [95%CI 13.3 to 17.0%]. Forty-four participants (1.6%, 95%CI: 1.2-2.1%) were positive for SARS-CoV-2 RT-PCR. IgG against SARS-CoV-2 was observed in 12.8% [95%CI: 11.6-14.1%] of participants.

(4) Overall, while waiting for population vaccination and/or increased herd immunity, we should concentrate in identifying and isolating new cases and their contacts.

2-          Sorne more keywords can be included.

Response: Following the reviewer instructions we introduced some keywords (i.e, “Keywords: Coronavirus; seroprevalence; SARS-CoV-2; infection status; university community; COVID-19 prevalence; faculty members; Spain; students; administrative and service staff”).

3-          Limitations of the study can be included in the revised work.

Response: We thank the reviewer for his/her comment. Thus, the limitations of the study have been already included in the discussion section (“Some limitations need to be considered for interpretation of the results of this cross-sectional study. Whilst we intended to obtain a representative sample of the university community, the low participation rate prompted to finally invite to participate all the target community, thus allowing to obtain a sufficient sample size. The final sample obtained was similar in several baseline characteristics, such as sex and age, to the overall university population, but not by community groups and schools. This might imply that generalization to the overall UB population or more general university populations must be done with caution. Finally, other potential limitations of the study should be considered, for instance the fact that the clinical background data shown in Table 1 and 2 is based on self-reporting by participants (i.e., not formally recorded by a clinician), thus this information should be considered accordingly, or that our questionnaire did not recorded information regarding the preventive measures taken by participants during their daily tasks”).

4-          In introduction section, limitations of the related studies should be discussed.

Response: As stated in the last reviewer query, we already highlighted the limitations of the study in the discussion section.

5-          Contribution can be marked with bullets or numbers in the revised work.

Response: All new contributions to the manuscript are marked in red in the revised work.

6-          It is better to mark the steps of the proposed concept working in Section 111 with a comprehensive figure in the revised work.

Response: We thank the reviewer for his/her constructive comments. We follow the STROBE Statement recommendations (Vandenbroucke et al. 2007 Ann Intern Med. 147: W-163–W-194) while reporting our observational studies. To this end, we included a Figure (Figure 1) which provides an overview of the flow chart of UB members involved in the study.

7-          How the results can be compared with sorne prior work?

Response: The reviewer pointed out an important question here. Indeed, we already compared our results at the national and international level (“The seroprevalence found (12.8 %) was greater than that estimated in Catalonia in December 2020: 9.2 % (95 %CI: 7,7 - 11,0), and by gender of 8,6% (95%CI: 6,9 - 10,5) in males and of 9,8% (95%CI: 8,1 - 11,8) in females [16]; and greater than the seroprevalence estimated in Spain in the same study [overall population of 7.1 % (95 %CI: 6.7 – 7.6) and by gender of 6.7 % (95 %CI: 6.2 – 7.2) in male and of 7,5% (95 %CI: 6.9 – 8.0) in female]. Interestingly, the results reported here are in line with these recently reported for the SARS-CoV-2 seroprevalence in 2,905 university students from five different universities within England (17.8%, 95%CI, 16.5-19.3), ranging between 7.6%-29.7% [26]”).

8-          In introduction section, the organization of this article can be included concisely with few sentences.

Response: We thank the reviewer for this comment. Again, we followed the journal’s instructions for the completion of the introduction (i.e., The introduction should briefly place the study in a broad context and highlight why it is important. It should define the purpose of the work and its significance, including specific hypotheses being tested. The current state of the research field should be reviewed carefully, and key publications cited. Please highlight controversial and diverging hypotheses when necessary. Finally, briefly mention the main aim of the work and highlight the main conclusions. Keep the introduction comprehensible to scientists working outside the topic of the paper). Interestingly, these instructions agree with the recommendation of the International Committee of Medical Journal Editors (ICJME) which do not consider such type of introductory statement.

9-          Future course of action can be included in the revised work.

Response: Thanks for the comment, we have introduced within the Discussion section some indications of implications for public health and COVID-19 control (i.e., Considering the global situation, including economic recession, and while we wait for most of the population to be vaccinated and/or for herd immunity to be achieved, our results should be interpreted as enough to keep the current nonpharmacologic preventive measures taken, such as mask-wearing, social distancing, hand hygiene (with hydroalcoholic gels), ventilated spaces, and “shelter-in-place” a minimum of 10 days in case of being a contact”).

10-        What are the important research implication of this study?

Response: Again, we thank the reviewer for his/her comments. This question is somehow related to the last one, thus we already commented the implications of this study in the Conclusion section (“While these results do not suffice to deescalate COVID-19 preventive measures (i.e., mask-wearing, social distancing, ventilated spaces) right now, the continuous growing of SARS-CoV-2 infection prevalence and the community vaccination will contribute to achieve a minimal herd immunity needed to start thinking in eventual and progressive measures to adapt the teaching and research activities towards the normality along the next academic year”).

11-        It would be better to include sorne more scenarios or result to prove the feasibility of the proposed method.

Response: We thank the reviewer for the comment. The method to establish the prevalence of any health condition consists in a cross-sectional study determining a variable(s). In or study we performed cross-sectional study conducted among UB members to determine the prevalence of SARS-Cov-2 infection. In our case, it seems difficult to foresee any other scenario to prove the feasibility of the proposed method.

12-        Careful proofreading of the article is paramount during the revisions.

Response: Indeed, we have proofread the manuscript carefully before resubmission.

Reviewer 3 Report

Thank you for submitting this manuscript on a very current topic. Please find below my detailed comments that may further strengthen the manuscript.

  • Please explain the abbreviations in the abstract.
  • Please re-order the introduction: starting with the broader topic and ending with the description of the university and the rationale
  • What is the rationale for including only university students and staff? The rationale has to be clearer.
  • line 98: What prevalence? Covid?
  • Please provide more details on the "epidemiological questionnaire" and the coding of the variables.
  • Is the study population representative for the entire population?
  • What was the result of the sample size calculation? Was the predefined sample size reached?

Author Response

Thank you for submitting this manuscript on a very current topic. Please find below my detailed comments that may further strengthen the manuscript.

  • Please explain the abbreviations in the abstract.

Response:

We apologize for the inconvenience. We now explained the abbreviations in the abstract.

  • Please re-order the introduction: starting with the broader topic and ending with the description of the university and the rationale

Response:

We thanks to the reviewer for his/her constructive comment. Thus, we rewrite the introduction following his/her instructions.

  • What is the rationale for including only university students and staff? The rationale has to be clearer.

Response:

The reviewer pointed out an important question here. We performed a cross-sectional study conducted among UB members stratified into four community groups: i) Students; ii) Administrative and service staff (ASS); iii) Faculty members (FM); and iv) Clinical faculty members (CFM) (see Table 1 and 2). The rationale of using these four community groups is that any member of the University of Barcelona will fall into any of the abovementioned groups, thus no member being left out of the study. Accordingly, we include this rationale in the new version of the manuscript (“Due to the heterogeneity of sociodemographic characteristics across the UB population, stratification was based on students, ASS and faculty members. This last group was also divided into clinical faculty and non-clinical faculty members (i.e., CFM and FM) due to an expected higher exposition to SARS-CoV-2 among the first. By using this four groups stratification no UB member was left out of the study”).

  • line 98: What prevalence? Covid?

Response:

We apologize for the inconvenience. We now indicated which prevalence refers to (“…an expected SARS-CoV-2 seroprevalence of 7.5%).

  • Please provide more details on the "epidemiological questionnaire" and the coding of the variables.

Response:

As the reviewer suggested, we provide more details on the epidemiological questionnaire (“Once the participant accepted online, the participants were required to answer a short online epidemiological questionnaire. This questionnaire gathered information about sociodemographic variables, self-reported clinical background (including estimated body masa index and COVID-19-related symptoms), lifestyle habits (i.e., tobacco and alcohol use), previous screening for SARS-CoV-2 (i.e., RT-PCR and/or serology) and risk of SARS-CoV-2 infection (i.e., contact with infected people)”.

  • Is the study population representative for the entire population?

Response: The reviewer highlighted and important issue here. We already discussed this matter in the discussion section (“Some limitations need to be considered for interpretation of the results of this cross-sectional study. Whilst we intended to obtain a representative sample of the university community, the low participation rate prompted to finally invite to participate all the target community, thus allowing to obtain a sufficient sample size. The final sample obtained was similar in several baseline characteristics, such as sex and age, to the overall university population, but not by community groups and schools. This might imply that generalization to the overall UB population or more general university pop-ulations must be done with caution).

  • What was the result of the sample size calculation? Was the predefined sample size reached?

Response: We thank the reviewer for pointing out this question. Again, this issue has been already discussed in the manuscript (“We estimated a sample size of 4,944 participants for an expected SARS-CoV-2 seroprevalence of 7.5% for students according to a nationwide, population-based seroepidemiological study (ENE-COVID Study) [16], and 12% or higher for clinical faculty. In anticipation, 30% of the non-response rate was accounted for. On December 9th, 2020, 4,944 UB members were randomly selected and invited to participate by e-mail. After three reminders, only 370 individuals participated in the study. On February 1st, 2021, after this initial attempt, a new sample of 4,944 UB members was randomly selected and invited to participate by e-mail with similar results. In view of this low participation rate, we decided to contact and invite to participate the remaining UB population in two other successive waves, on 8th February (n=26,671) and 15th of February 2021 (n=15,961)”.

Round 2

Reviewer 2 Report

Dear Authors, the paper is well organized and clear now. My all  major concerns have been addressed. In my opinion, the paper can be further improved by incorporating following minor suggestions. I acknowledge and congratulate the authors for their significant efforts and the time they spent on the revision of the Manuscript.

  • I would recommend having a second look on the manuscript title, I think the key utility of the method in COVID context must be highlighted. For example, ‘Analysis of SARS-CoV-2 infection Dynamics at the University of Barcelona during the third wave COVID-19 pandemic wave’. Authors can retain the same title too.
  • Future course of action can be included concisely.

Author Response

We thank the reviewer for the careful assessment of the manuscript. Indeed, the title would be possible to readjust. However, we believe that introducing the word “Dynamics” might generate confusion as here we performed a one-off or cross-sectional study (i.e., a single picture of the COVID-19 situation at the University of Barcelona) rather that a longitudinal study following the same sample of people over time, which may have a dynamic connotation. Therefore, we would like to keep the title as it is right now.

Finally, we have introduced within the Discussion section some future course of action or indications for COVID-19 control (i.e., Considering the global situation, including economic recession, and while we wait for most of the population to be vaccinated and/or for herd immunity to be achieved, our results should be interpreted as enough to keep the current nonpharmacologic preventive measures taken, such as mask-wearing, social distancing, hand hygiene (with hydroalcoholic gels), ventilated spaces, and “shelter-in-place” a minimum of 10 days in case of being a contact”).

Reviewer 3 Report

I would like to thank the authors for revising the manuscript. Although the manuscript has improved, I have some further comments:

Please include more information on the questionnaire: The reader would like to know the exact variables and the response categories in the methods section. Although this section has been revised, the new content is very superficial and needs more detailed information.

Did I understand correctly, that the sample calculation showed that 4,944 individuals were needed and in the first step only 4,944 individuals were invited? To reach the calculated sample size, a response of 100% would have been necessary. This presumption is a weakness that should be named in the limitation section.

Since data are only drawn from one university, and the basic characteristic of the overall university population is known, study data should be weighted. From my point of view, it is not sufficient to describe that there is a difference between the sample and the basic population, when authors have the opportunity to calculate weights. This would strengthen the entire manuscript, because the authors could report data, that is "representative" for a single university.

Author Response

I would like to thank the authors for revising the manuscript. Although the manuscript has improved, I have some further comments:

Please include more information on the questionnaire: The reader would like to know the exact variables and the response categories in the methods section. Although this section has been revised, the new content is very superficial and needs more detailed information.

Response:

We apologize for the inconvenience. The reviewer is right, more information regarding the questionnaire should be provided. Accordingly, we now provide the questionnaire used in the study (see Appendix B) showing all the variables considered.

Did I understand correctly, that the sample calculation showed that 4,944 individuals were needed and in the first step only 4,944 individuals were invited? To reach the calculated sample size, a response of 100% would have been necessary. This presumption is a weakness that should be named in the limitation section.

Response:

We apologize for the misunderstanding here. We now clarify this part in the new version of the manuscript (“This was a cross-sectional study conducted among UB members between Decem-ber 14th, 2020 and February 25th, 2021. According to a nationwide, population-based seroepidemiological study (ENE-COVID Study) [16] we expected a SARS-CoV-2 sero-prevalence of 7.5% for students and 12%, or higher, for clinical faculty members. In ad-dition, we assumed a 30% of non-response rate. Thus, to reach a final overall sample size of 3,450 individuals we estimated an initial sample size of 4,944 participants to be con-tacted. On December 9th, 2020, 4,944 UB members were randomly selected and invited to participate by e-mail. After three reminders, only 370 individuals participated in the study, thus providing rate of participation of 7.5 % (almost ten-times less to the expected one). Consequently, on February 1st, 2021, after this initial attempt, a new sample of 4,944 UB members was randomly selected and invited to participate by e-mail with similar results. Finally, in view of this low participation rate, we decided to contact and invite to participate the remaining UB population in two other successive waves, on 8th February (n=26,671) and 15th of February 2021 (n=15,961). A final study population of 2,784 par-ticipants was considered (Figure 1). UB members were contacted by the information of the most recent census (updated for the UB President’s office election in December 2020).

Since data are only drawn from one university, and the basic characteristic of the overall university population is known, study data should be weighted. From my point of view, it is not sufficient to describe that there is a difference between the sample and the basic population, when authors have the opportunity to calculate weights. This would strengthen the entire manuscript, because the authors could report data, that is "representative" for a single university.

The reviewer highlighted an important issue here. In our study sample weights were used to estimate and report the global prevalence, as shown in the statistical analysis section "Global RT-PCR-positive prevalence and global seroprevalence were estimated using sampling weights", and footnote on figure 2 "SARS-CoV-2 infection prevalence (assayed by RT-PCR and IgG serology) estimated using sampling weights by UB community groups".